# Parameter-free Dynamic Graph Embedding for Link Prediction

**Jiahao Liu[1,2], Dongsheng Li[3], Hansu Gu[4✉], Tun Lu[1,2✉], Peng Zhang[1,2], Ning Gu[1,2]**

[1]School of Computer Science, Fudan University, Shanghai, China
[2]Shanghai Key Laboratory of Data Science, Fudan University, Shanghai, China
[3]Microsoft Research Asia, Shanghai, China    [4]Seattle, United States
jiahaoliu21@m.fudan.edu.cn, dongsli@microsoft.com, hansug@acm.org
{lutun, zhangpeng_, ninggu}@fudan.edu.cn

## Abstract

Dynamic interaction graphs have been widely adopted to model the evolution of user-item interactions over time. There are two crucial factors when modelling user preferences for link prediction in dynamic interaction graphs: 1) collaborative relationship among users and 2) user personalized interaction patterns. Existing methods often implicitly consider these two factors together, which may lead to noisy user modelling when the two factors diverge. In addition, they usually require time-consuming parameter learning with back-propagation, which is prohibitive for real-time user preference modelling. To this end, this paper proposes FreeGEM, a parameter-free dynamic graph embedding method for link prediction. Firstly, to take advantage of the collaborative relationships, we propose an incremental graph embedding engine to obtain user/item embeddings, which is an Online-Monitor-Offline architecture consisting of an Online module to approximately embed users/items over time, a Monitor module to estimate the approximation error in real time and an Offline module to calibrate the user/item embeddings when the online approximation errors exceed a threshold. Meanwhile, we integrate attribute information into the model, which enables FreeGEM to better model users belonging to some under represented groups. Secondly, we design a personalized dynamic interaction pattern modeller, which combines dynamic time decay with attention mechanism to model user short-term interests. Experimental results on two link prediction tasks show that FreeGEM can outperform the state-of-the-art methods in accuracy while achieving over 36X improvement in efficiency. All code and datasets can be found in https://github.com/FudanCISL/FreeGEM.

## 1 Introduction

Dynamic interaction graphs have been adopted in a wide range of link prediction tasks to model the evolution of user-item interactions over time [12, 14, 37]. The evolution of a dynamic interaction graph can be reflected by its historical interaction (edge) sequence, in which two crucial factors should be explicitly considered for link prediction tasks: 1) collaborative relationship among users, i.e., users with similar historical interactions will interact with similar items in the future, which is also the basic assumption of collaborative filtering [7, 15, 20]; and 2) user personalized dynamic interaction patterns, i.e., users could have unique short-term interaction patterns, which are not shared among like-minded users but can only be reflected by their own interaction sequences.

Many existing methods learn user preferences from dynamic interaction graphs using temporal point process (TPP) [28, 30, 41], recurrent neural network (RNN) [1, 14, 32] and graph neural network (GNN) [4, 23, 37], etc., in which the following key challenges arise. Firstly, these methods do not

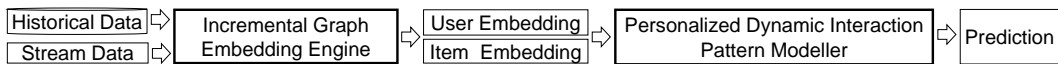

Figure 1: The high-level design of the proposed FreeGEM method.

consider the above two key factors separately, so that collaborative relationship may bring noises to personalized patterns when they diverge, and vice versa. Secondly, the above methods often require time-consuming parameter learning with back-propagation. In dynamic interaction graphs, the model training should follow chronological order of the interactions to capture the temporal dynamics, which raises efficiency issue even for applications with moderate number of interactions.

In this paper, we propose a Parameter-**Free** Dynamic **G**raph **EM**bedding (FreeGEM) method for link prediction. *Here, parameter-free means that we do not incorporate any parameters which need to be learned via back-propagation.* As shown in Figure 1, FreeGEM consists of two key components: incremental graph embedding engine and personalized dynamic interaction pattern modeller. The incremental graph embedding engine takes both historical data and online stream data as input and generates user/item embeddings as the output, which exploits the collaborative relationship among users. Its core innovation is the proposed Online-Monitor-Offline architecture, which can achieve online embedding updates by solving closed-form solutions in real time and keep the approximation errors caused by online singular value decomposition (SVD) [2] within any predefined threshold. In the Offline step, we propose a frequency-aware preference matrix reconstruction method to alleviate the oversmoothing problem and an attribute-integrated SVD to alleviate the cold-start issue. Surprisingly, the integration of attribute information enables FreeGEM to better model users belonging to under represented groups. In the Online step, we convert the offline truncated SVD into an online SVD to generate embeddings in real time. In the Monitor module, we estimate the online approximation error in real time by analyzing the relationship between approximation error and the update of online algorithm. The personalized dynamic interaction pattern modeller is also a parameter-free component, which combines the dynamic time decay with attention mechanism to model user short-term interests. It takes the user and item embeddings as input and outputs the prediction results. Specifically, it selectively forgets the early interactions through dynamic time decay and hence focuses on more recent interactions for prediction. Then, it leverages the attention mechanism to capture user personalized dynamic interaction patterns over the "decayed" item embedding sequences. Experimental results on two link prediction tasks (future item recommendation and next interaction prediction) show that FreeGEM can substantially outperform the state-of-the-art link prediction methods in accuracy while achieving over 36X improvement in computational efficiency. Besides, our empirical studies also confirm that FreeGEM can alleviate the cold-start issue and achieve high robustness on very sparse data.

## 2   Related Work

Link prediction methods on dynamic interaction graphs are mostly based on TPP, RNN and GNN, etc., which need time-consuming training process. To the best of our knowledge, the proposed FreeGEM is the only parameter-free method in this task, with high accuracy and efficiency.

**TPP-based methods**   Know-Evolve [28] and HTNE [41] model interactions as multivariate point processes and Hawkes processes, respectively. Wang et al. [30] propose a co-evolutionary process to model the co-evolving nature of users and items. Shchur et al. [22] propose to directly model the conditional distribution of inter-event times. DSPP [3] incorporates topology and long-term dependencies into the intensity function.

**RNN-based methods**   RRN [32] models user and item interaction sequences with separate RNNs. Time-LSTM [40] proposes time gates to represent the time intervals. LatentCross [1] incorporates contextual data into embeddings. DeepCoevolve [6] and JODIE [14] generate node embeddings using two intertwined RNNs. JODIE [14] can also estimate user embeddings trajectories. DeePRed [13] employs non-recursive mutual RNNs to model interactions.

**GNN-based methods**   TDIG-MPNN [4] captures the global and local information on the graph. DGCF [16] uses three update mechanisms to update users and items. SDGNN [27] takes the

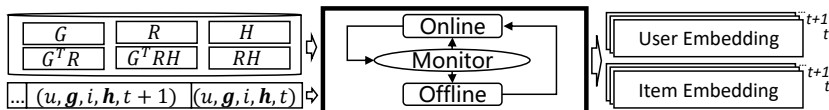

Figure 2: Illustration of the proposed incremental graph embedding engine.

state changes of neighbor nodes into account. MRATE [5] combines different relations to realize multi-relation awareness. OGN [11] updates nodes in an online fashion with a constant memory cost. TCL [29] proposes a graph-topology-aware transformer to learn the representations of nodes. CoPE [37] uses an ordinary differential equation-based GNN to model the evolution of network. TGL [39] proposes a unified framework for large-scale temporal GNN training. MetaDyGNN [34] proposes a model based on a meta-learning framework for few-shot link prediction in dynamic networks. TREND [31] uses Hawkes process-based GNN for temporal graph representation learning.

## 3 Incremental Graph Embedding Engine

As shown in Figure 2, the incremental graph embedding engine 1) takes the historical user-item interaction, user/item attribute and online stream data as the input, 2) embeds users and items via the novel Online-Monitor-Offline architecture, and 3) outputs the user and item embeddings in real time.

**Online-Monitor-Offline Architecture**   The Offline module first decomposes the historical data matrices by offline SVD to obtain user and item embeddings. To capture real-time information from the interaction stream, the model needs to efficiently update the corresponding embeddings when new interaction occurs. Thus, we propose the Online module, which uses online SVD to approximately update user and item embeddings, to meet the real-time requirements. However, since the approximation error of online SVD accumulates over time, we propose a Monitor module to estimate the accumulated approximation error of the Online module in real time. When the accumulated error exceeds a threshold, we restart the Offline module to calibrate the user and item embeddings. Otherwise, we continue to execute the Online module for real-time user/item embedding.

### 3.1 Offline Module

The input data of FreeGEM includes six matrices as defined in Table 1, in which $R \in \mathbb{R}^{m \times n}$ is user-item interaction matrix, $G \in \mathbb{R}^{m \times p}$ is user attribute matrix and $H \in \mathbb{R}^{n \times q}$ is item attribute matrix, where $m/n$ is the number of users/items, $q/p$ is the dimension of user/item attribute vector and $(R)_{ij}$ indicates the number of interactions between user $i$ and item $j$. Each stream data sample can be represented as a 5-tuple $(u, \mathbf{g}, i, \mathbf{h}, t)$, where $u$, $i$, $t$ are user id, item id and timestamp respectively, and $\mathbf{g} \in \mathbb{R}^p$ and $\mathbf{h} \in \mathbb{R}^q$ are user and item attribute vectors, respectively.

#### 3.1.1 Frequency-aware Preference Matrix Reconstruction

There are three steps to realize collaborative filtering [19]: 1) decompose the interaction matrix $R$ using truncated SVD to obtain $U = (\mathbf{u}_1, ..., \mathbf{u}_k) \in \mathbb{R}^{m \times k}$, $S = diag(s_1, ..., s_k) \in \mathbb{R}^{k \times k}$, $s_1 > ... > s_k > 0$ and $V = (\mathbf{v}_1, ..., \mathbf{v}_k) \in \mathbb{R}^{n \times k}$; 2) the embeddings of users and items are $E_U = US^{1/2}$ and $E_I = VS^{1/2}$, respectively; and 3) use $\hat{R} = E_U E_I^\top$ as the low rank approximation of $R$. Our method makes non-trivial improvements based on this, including frequency-aware preference matrix reconstruction in this section and attribute-integrated SVD in the Section 3.1.2.

The raw interaction matrix $R$ cannot perfectly reflect user preferences due to potential biases [24, 25], so we propose to normalize $R$ as follows: $R' = D_U^{-\alpha} R D_I^{-\alpha}$, where $\alpha > 0$ is a hyperparameter, $D_U$ and $D_I$ are diagonal matrices, and $(D_U)_{ii} = \Sigma_{j=1}^n R_{ij}$ and $(D_I)_{jj} = \Sigma_{i=1}^m R_{ij}$. Compared with $R$, $R'$ can more accurately represent user preferences. Intuitively, the more users an item interacts with, the less it can reflect each user's preference, and vice versa. The normalization can alleviate the popular bias and help to achieve more accurate user representation.

By applying offline truncated SVD on $R'$, we have the approximation of $R'$ as: $\hat{R}' = \Sigma_{i=1}^k s_i \mathbf{u}_i \mathbf{v}_i^\top$, which only retains the low-frequency signals of $R'$, so it can be regarded as $R'$ passing through an

Table 1: Correspondence between the decomposed matrix and the constructed embedded matrix.

| Description of the decomposed matrix | Decomposed matrix | Constructed embedding (left) | Constructed embedding (right) |
|---|---|---|---|
| user-item | $R \in \mathbb{R}^{m \times n}$ | $E_U^1 \in \mathbb{R}^{m \times k_1}$ | $E_I^1 \in \mathbb{R}^{n \times k_1}$ |
| user attribute | $G \in \mathbb{R}^{m \times p}$ | $E_U^2 \in \mathbb{R}^{m \times k_2}$ | $E_G^2 \in \mathbb{R}^{p \times k_2}$ |
| item attribute | $H \in \mathbb{R}^{n \times q}$ | $E_I^3 \in \mathbb{R}^{n \times k_3}$ | $E_H^3 \in \mathbb{R}^{q \times k_3}$ |
| user_attribute-item | $G^\top R \in \mathbb{R}^{p \times n}$ | $E_G^4 \in \mathbb{R}^{p \times k_4}$ | $E_I^4 \in \mathbb{R}^{n \times k_4}$ |
| user-item_attribute | $RH \in \mathbb{R}^{m \times q}$ | $E_U^5 \in \mathbb{R}^{m \times k_5}$ | $E_H^5 \in \mathbb{R}^{q \times k_5}$ |
| user_attribute-item_attribute | $G^\top RH \in \mathbb{R}^{p \times q}$ | $E_G^6 \in \mathbb{R}^{p \times k_6}$ | $E_H^6 \in \mathbb{R}^{q \times k_6}$ |

Table 2: Correspondence between path and embedding matrix.

| No. | Path | User embedding | Item embedding |
|---|---|---|---|
| 1 | user-item | $E_{U_1} = E_U^1 \in \mathbb{R}^{m \times k_1}$ | $E_{I_1} = E_I^1 \in \mathbb{R}^{n \times k_1}$ |
| 2 | user-user_attribute-item | $E_{U_2} = E_U^2 (E_G^2)^\top \in \mathbb{R}^{m \times p}$ | $E_{I_2} = E_I^4 (E_G^4)^\top \in \mathbb{R}^{n \times p}$ |
| 3 | user-item_attribute-item | $E_{U_3} = E_U^5 (E_H^5)^\top \in \mathbb{R}^{m \times q}$ | $E_{I_3} = E_I^3 (E_H^3)^\top \in \mathbb{R}^{n \times q}$ |
| 4 | user-user_attribute-item_attribute-item | $E_{U_4} = E_U^2 (E_G^2)^\top E_G^6 \in \mathbb{R}^{m \times k_6}$ | $E_{I_4} = E_I^3 (E_H^3)^\top E_H^6 \in \mathbb{R}^{n \times k_6}$ |
| 5 | user-item_attribute-user_attribute-item | $E_{U_5} = E_U^5 (E_H^5)^\top E_H^6 \in \mathbb{R}^{m \times k_6}$ | $E_{I_5} = E_I^4 (E_G^4)^\top E_G^6 \in \mathbb{R}^{n \times k_6}$ |

ideal low-pass graph filter. Thus, it will suffer from the oversmoothing problem when $k$ is small. To alleviate this problem, we introduce a hyperparameter $\gamma$ to control the ratio of high-frequency signals to low-frequency signals as follows: $E_U = US^\gamma$, $E_I = VS^\gamma$ ($\gamma < 0.5$). The original intensity of each frequency is $s_i$, which becomes $s_i^{2\gamma}$ after applying the frequency control. The attenuation ratio is $s_i^{2\gamma}/s_i$, which is a hyperbola about $s_i$. The low-frequency signal corresponds to a larger $s_i$, which means that the attenuation of the low-frequency signal is stronger than that of the high-frequency signal, improving the proportion of the high-frequency signal in the reconstructed matrix $\hat{R}'$. In summary, this is equivalent to reducing the proportion of low-frequency signals after ideal low-pass filtering through truncated SVD (more discussion can be found in the Appendix A.1). Finally, the predicted interaction matrix is obtained by inverse normalization as follows: $\hat{R} = D_U^\alpha \hat{R}' D_I^\alpha$.

### 3.1.2 Attribute-integrated SVD

In addition to frequency-aware preference matrix reconstruction, compared with the plain SVD, we also integrate attribute information to better model users and items. Integrating user and item attributes can help to improve the prediction accuracy and alleviate the cold-start issue in the recommender system [33, 36]. $R$, $G$ and $H$ are three basic matrices, which respectively describe the co-occurrence relationship of user-item, user-user_attribute and item-item_attribute. To connect them, we further obtain three derived matrices $G^\top RH \in \mathbb{R}^{p \times q}$, $G^\top R \in \mathbb{R}^{p \times n}$ and $RH \in \mathbb{R}^{m \times q}$, which describe the co-occurrence relationship of user_attribute-item_attribute, user_attribute-item and user-item_attribute respectively. After decomposing and reconstructing the six matrices using the method described in Section 3.1.1, we can get $6 \times 2 = 12$ embedding matrices, as shown in Table 1, whose superscript indicate embedding space, corresponding to different dimensions. As shown in Figure 3, we have four objects: user, user attribute, item and item attribute. There are $C_4^2 = 6$ relationships between them, which are represented by the six matrices respectively. The six edges represent six embedding spaces, and each object has a representation in three of them. There are five paths between user and item without revisiting, each of which corresponds to a user embedding matrix and an item embedding matrix, as shown in Table 2. Except for the user-item path where user and item are directly connected, all other paths go through user/item attribute and thus integrate the user/item attributes in the embeddings. Finally, we only concatenate the embeddings from the first three paths as the output of the Offline module, for the last two paths consider user/item attributes repeatedly, and obtain user embedding and item embedding as follows:

$$E_U = (\alpha_1 E_{U_1})||(\alpha_2 E_{U_2})||(\alpha_3 E_{U_3}) \in \mathbb{R}^{m \times (k_1 + p + q)}. \tag{1}$$

$$E_I = (\alpha_1 E_{I_1})||(\alpha_2 E_{I_2})||(\alpha_3 E_{I_3}) \in \mathbb{R}^{n \times (k_1 + p + q)}. \tag{2}$$

$\alpha_i$ is the hyperparameter that controls the weight of the $i$-th path, and $||$ represents concatenation.

### 3.2 Online Module

The only difference between Online module and Offline module is the way to obtain truncated SVD, where the Offline module uses ordinary truncated SVD but the Online module uses online SVD [2]. Online SVD [2] provides an approximated method to calculate the truncated SVD of an updated

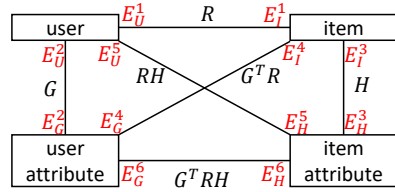

Figure 3: Attribute-integrated SVD.

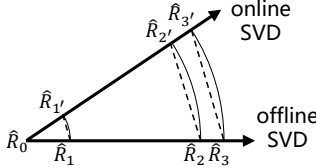

Figure 4: Motivating example of the Monitor.

matrix in linear time. Let $R_t \in \mathbb{R}^{m \times n}$ be the interaction matrix at time $t$. Assuming that we have finished the truncated SVD of $R_t$, and then at time $t + 1$, user $u$ interacts with item $i$. Thus, we have $R_{t+1} = R_t + \mathbf{u}\mathbf{i}^\top$, where $\mathbf{u} \in \mathbb{R}^m$ and $\mathbf{i} \in \mathbb{R}^n$ are one-hot vectors indicating user id and item id, respectively. Therefore, we realize the incremental calculation of truncated SVD of $R_{t+1}$ given the truncated SVD of $R_t = USV^\top$, where $U \in \mathbb{R}^{m \times k}$, $S \in \mathbb{R}^{k \times k}$, $V \in \mathbb{R}^{n \times k}$ and $k = k_1 + p + q$ as shown in Equation 1 and Equation 2. The details are described as follows.

Firstly, we calculate:

$$\mathbf{m} = U^\top \mathbf{u}, \ \mathbf{n} = V^\top \mathbf{i}, \ \mathbf{p} = \mathbf{u} - U\mathbf{m}, \ \mathbf{q} = \mathbf{i} - V\mathbf{n}, \ P = ||\mathbf{p}||^{-1}\mathbf{p}, \ Q = ||\mathbf{q}||^{-1}\mathbf{q}. \quad (3)$$

Secondly, we calculate:

$$K = \begin{bmatrix} S & \mathbf{0} \\ \mathbf{0} & 0 \end{bmatrix} + \begin{bmatrix} \mathbf{m} \\ ||\mathbf{p}|| \end{bmatrix} \begin{bmatrix} \mathbf{n} \\ ||\mathbf{q}|| \end{bmatrix}^\top. \quad (4)$$

Thirdly, we calculate the full SVD of $K$ and get $U_K S_K V_K^\top$ with $U_K \in \mathbb{R}^{(k+1) \times (k+1)}$, $V_K \in \mathbb{R}^{(k+1) \times (k+1)}$ and $S_K \in \mathbb{R}^{(k+1) \times (k+1)}$. Then, we have the following result:

$$R_{t+1} = R_t + \mathbf{u}\mathbf{i}^\top \approx ([U \ P]U_K)S_K([V \ Q]V_K))^\top. \quad (5)$$

Finally, we can use the first $k$ columns of $[U \ P]U_K$, $S_K$ and $[V \ Q]V_K$ as the estimate of the truncated SVD of $R_{t+1}$.

### 3.3 Monitor Module

In the Online module, the approximation error will accumulate over time. To analyze the accumulated error, TIMERS [38] was proposed to calculate the lower bound of the approximation error of online SVD through matrix perturbation [26]. However, TIMERS depends on a time-consuming eigen-decomposition, so it can not monitor the error in real time but based on the granularity of time slice for eigen-decomposition. Nevertheless, TIMERS provides two insights for the design of the Monitor: 1) the accumulated approximation error does not change uniformly with time and 2) there are no shared correlation patterns between the running time of online SVD, the number of new observations in the interaction matrices and the accumulated error among different datasets.

The timing of restart Offline module is very important because untimely restart will lead to excessive error and reduce the embedding quality and too frequent restart will lead to low efficiency. There could be two restart heuristics: 1) restart after a certain time and 2) restart after a certain number of new interactions. However, both heuristics are not optimal due to unawareness of the online approximation error, leading to potentially unnecessary or inaccurate updates.

The motivation of the Monitor is illustrated in Figure 4. At time step 0, we initialize the offline truncated SVD of the user-item interaction matrix $\hat{R}_0$. At time step $i$, we can have matrices: 1) $\hat{R}_i$ if we use offline SVD and 2) $\hat{R}_{i'}$ if we use online SVD to approximate the matrix when new interaction occurs. For ease of illustration, we use straight lines to represent the F-norm distance of the reconstructed matrices, and arrows to represent the evolution directions of the results. The angle between the evolution directions of offline SVD and online SVD should be less than $\pi/3$ (more discussion can be found in the Appendix A.2), otherwise the results of online SVD is even worse than not updating at all. Since there is no online approximation error in offline SVD, we take $\hat{R}_i$, where $i = 1, 2, 3$, as the ground truth and $\hat{R}_{i'}$, where $i = 1, 2, 3$, correspond to the results obtained by online SVD after each update. The lengths of $\hat{R}_0\hat{R}_1$ ($\hat{R}_0\hat{R}_{1'}$), $\hat{R}_1\hat{R}_2$ ($\hat{R}_{1'}\hat{R}_{2'}$) and $\hat{R}_2\hat{R}_3$ ($\hat{R}_{2'}\hat{R}_{3'}$)

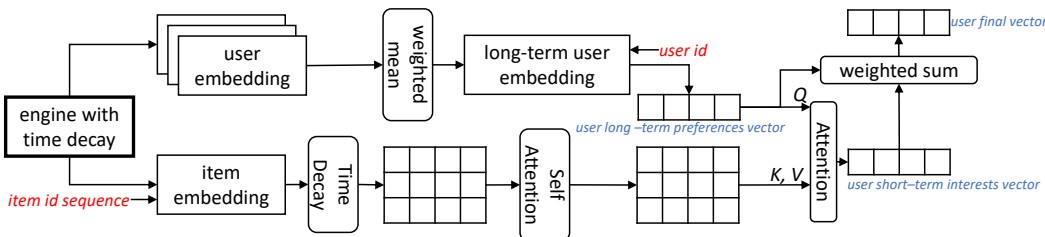

Figure 5: Personalized dynamic interaction pattern modeling module.

are not equal, for different interactions have different impacts on the evolution of SVD [38]. The dotted line represents the F-norm distance between the matrices reconstructed by offline SVD and online SVD, $e = ||\hat{R}_i - \hat{R}_{i'}||_F$, i.e., the online approximation error. The dotted line become longer over time, indicating that the error will accumulate with the occurrence of new interactions.

We use *distance* to specifically refer to the F-norm distance between the reconstructed matrix by online SVD at the current time step and the initial reconstructed matrix: $d = ||\hat{R}_{i'} - \hat{R}_0||_F$. Calculating $\hat{R}_1$, $\hat{R}_2$, $\hat{R}_3$ is time-consuming, so we hope to monitor the length of the dotted line without calculating it. While calculating $\hat{R}_{1'}$, $\hat{R}_{2'}$, $\hat{R}_{3'}$ is efficient due to the linearity of online SVD and it can be seen from the Figure 4 that there is a positive correlation between *distance* and approximation error (empirically verified in the Appendix A.3), so we can estimate the online approximation error though the *distance*. Every time the Online module is executed, the Monitor will calculate the *distance* in real time. When the *distance* exceeds the predefined threshold, we believe the corresponding approximation error also reaches another threshold for restarting the Offline module.

Compared with the two simple heuristics, our Monitor estimates errors in a data-driven way with the following benefits: 1) the Monitor estimates the error according to the interaction events, avoiding the negative impacts caused by uneven interaction distributions in different time intervals; and 2) the Monitor estimates the error according to the positive correlation between the *distance* and the error, avoiding the varying impacts of different interactions on the error estimation. It should be noted that, compared with the two heuristics, Monitor only adds the process of calculating the *distance*, which is very efficient and thus can achieve real-time monitoring.

## 4 Personalized Dynamic Interaction Pattern Modeller

Both Offline module and Online module cannot model the evolution trends of the dynamic interaction graph due to lacking the ability of memorization. The incremental graph embedding engine captures collaborative relationships shared among similar users, but the interaction pattern of each user in his/her interaction sequence is irrelevant to his/her preference, i.e., interaction patterns are unique instead of collaborative with like-minded users. Therefore, we propose an additional downstream module to capture user personalized dynamic interaction patterns as illustrated in Figure 5.

### 4.1 Dynamic Time Decay

We define the process from the execution of an Offline module to the execution of the next Offline module a *stage*. At the beginning of the $i$-th stage, when constructing the historical interaction matrix, we decay the historical interaction score as $\exp\{\beta(t/T_i - 1)\}$, where $T_i$ is the beginning time of the $i$-th stage and $\beta > 0$ is the decay coefficient. This implies that, for each stage, the score of historical interaction ($t < T_i$) is less than 1, while the score of interactions in current stage ($t \geq T_i$) is greater than or equal to 1. The decay should be memoryless, i.e., the decay factor is unchanged for the same time difference (more discussion can be found in the Appendix A.4). This requires that $\beta/T$ is a constant. Thus, for stage $i$ and stage $j$, their decay coefficients have the following relationship: $\beta_i/\beta_j = T_i/T_j$.

Time decay makes the model more relevant to recent interactions through selective forgetting. It can help the model capture the interaction pattern that users often interact with the recently interacted items in many link prediction tasks. However, it cannot automatically capture the personalized

dynamic interaction pattern of each user. Thus, we propose to combine attention mechanism with dynamic time decay to achieve personalized interaction pattern modelling.

## 4.2 Attention Module

For a user, we use the weighted average of $a$ recent user embeddings as his/her long-term embedding:

$$\mathbf{e}_{long} = \sum_{r=1}^{a} \frac{1}{r}\mathbf{e}_u^{(r)}, \tag{6}$$

where $\mathbf{e}_u^{(r)} \in \mathbb{R}^k$ is the corresponding user embedding, that is, a row of the $E_U$, at the $r$-th time step. Then, we concatenate the decayed embeddings of the user's recent $b$ items as $S_u \in \mathbb{R}^{k \times b}$ and apply self-attention and attention to obtain the short term preference embedding $\mathbf{e}_{short}$ as follows:

$$S_u = \mathbf{e}_i^{(1)}||\cdots||\mathbf{e}_i^{(b)}, \;\; S_u' = \frac{S_u^\top S_u}{\sqrt{k}}S_u^\top, \;\; \mathbf{e}_{short} = (S_u')^\top \frac{S_u'\mathbf{e}_{long}}{\sqrt{k}}. \tag{7}$$

Finally, we obtain the final user embedding by a weighted average:

$$\mathbf{e} = \lambda\mathbf{e}_{short} + (1 - \lambda)\mathbf{e}_{long}. \tag{8}$$

We use the dot product between current item embeddings and user final embedding: $E_I \cdot \mathbf{e}$ as the scores for predicting the link between the user and all items. It should be noted that both self-attention and attention operations in our design require no weight matrices, i.e., they are parameter-free. The whole process is shown in the Figure 5.

**Discussion** A user has his/her own historical interactions, corresponding to personalization, and the recent historical interactions change over time, corresponding to temporal dynamic. A user's long-term preference vector should be changing smoothly, which is in line with the characteristics of long-term interests. A user's short-term preference vector is a linear combination of the embeddings of recently interacted items, which changes rapidly over time and is in line with the characteristics of short-term interests. Time decay of item sequences plays a key role in position embedding, which indicates that we will pay more attention to the recently interacting items. Self-attention can integrate the information of the whole sequence, and attention makes items with high similarity to users more likely to be interacted again. Compared with time decay, the attention module is data-driven and can automatically adapt to the personalized and dynamic interaction patterns by making full use of the recent interactions. However, compared with time decay, attention may be disturbed by many unnecessary patterns. Our empirical studies confirm that time decay and attention are complementary to each other and should be combined in the model.

## 5 Experiments

**Datasets** For the future item recommendation task, we use the Amazon Video, Amazon Game [9], MovieLens-1M (ML-1M) and MovieLens-100K (ML-100K) [8]. For the next interaction prediction task, we use Wikipedia and LastFM [14]. For all datasets, we use the first 80% interactions as training set, the following 10% interactions as validation set, and the last 10% interactions as test set.

**Metrics** For the future item recommendation task, we use *Recall@10* to evaluate models. For the next interaction prediction task, we use *MRR* and *Hit@10* to evaluate models. For all experiments, we report the results on the test set when the models achieve the optimal results on verification sets.

**Baselines** For the future item recommendation task, we compare with LightGCN [10], Time-LSTM [40], RRN [32], DeepCoevolve [6], JODIE [14] and CoPE [37]. For the next interaction prediction task, we compare with Time-LSTM [40], RRN [32], LatentCross [1], CTDNE [17], DeepCoevolve [6], JODIE [14] and CoPE [37]. Among all baselines, LightGCN [10] is the only static graph representation learning method for recommendation tasks. These baselines include TPP-based, RNN-based and GNN-based methods. For clarity, we discuss and compare these methods with FreeGEM in the Appendix A.5.

Table 3: Accuracy comparison with state-of-the-art methods on two link prediction tasks.

(a) Future item recommendation

| | Video | Game | ML-100K | ML-1M |
|---|---|---|---|---|
| | Recall | Recall | Recall | Recall |
| LightGCN | 0.036 | 0.026 | 0.025 | 0.029 |
| Time-LSTM | 0.044 | 0.020 | 0.058 | 0.033 |
| RRN | 0.068 | 0.029 | 0.065 | 0.043 |
| DeepCoevolve | 0.050 | 0.027 | 0.069 | 0.030 |
| JODIE* | 0.078 | 0.035 | 0.074 | 0.035 |
| CoPE* | 0.088 | 0.047 | 0.081 | 0.049 |
| **FreeGEM *(no attr)** | **0.113** | **0.050** | **0.114** | **0.053** |
| **FreeGEM *(with attr)** | - | - | **0.149** | **0.065** |

(b) Next interaction prediction

| | Wikipedia | | LastFM | |
|---|---|---|---|---|
| | MRR | Hit | MRR | Hit |
| Time-LSTM | 0.247 | 0.342 | 0.068 | 0.137 |
| RRN | 0.522 | 0.617 | 0.089 | 0.182 |
| LatentCross | 0.424 | 0.481 | 0.148 | 0.227 |
| CTDNE | 0.035 | 0.056 | 0.010 | 0.010 |
| DeepCoevolve | 0.515 | 0.563 | 0.019 | 0.039 |
| JODIE | 0.746 | 0.822 | 0.195 | 0.307 |
| CoPE | 0.750 | **0.890** | **0.200** | 0.446 |
| **FreeGEM** | **0.786** | 0.852 | 0.195 | **0.453** |

Table 4: The ablation studies on the future item recommendation task.

| | Pattern Modeller | Matrix Reconstruction | Video | Game | ML-100K | ML-1M |
|---|---|---|---|---|---|---|
| A | ✗ | ✗ | 0.073 | 0.023 | 0.051 | 0.045 |
| B | ✗ | ✓ | 0.107 | 0.041 | 0.074 | 0.051 |
| C | ✓ | ✗ | 0.080 | 0.025 | 0.113 | 0.052 |
| **FreeGEM *(no attr)** | ✓ | ✓ | 0.113 | 0.050 | 0.114 | 0.053 |

## 5.1 Future item recommendation

We use this task to verify whether the model can accurately predict user future interactions according to their historical interactions, which is a typical application of dynamic interaction graphs in recommender system. In this task, a user interacts with an item only once at most.

The results are shown in Table 3(a), in which the results of baselines are reported by CoPE [37]. To be fair, we do not allow JODIE, CoPE and FreeGEM to update models during test phase (marked with *). In addition, since all baselines do not integrate attribute, we provide both the results of FreeGEM with (*with-attr*) and without (*no-attr*) attributes. It can be observed that FreeGEM *(no-attr) achieves better accuracy than all baselines on all datasets. Compared with truncated SVD, FreeGEM *(no-attr) only adds the proposed frequency-aware reconstruction module and the personalized interaction pattern modeller. Further, FreeGEM *(with-attr) outperforms FreeGEM *(no-attr) due to the proposed attribute-integrated SVD, which verifies the effectiveness of this module. Among all baselines, LightGCN performs the worst, because it is the only static GNN model which cannot capture the dynamic characteristics of the interaction graph.

## 5.2 Next interaction prediction

We use this task to verify whether the model can accurately predict a user's next interaction according to the user's historical interactions up to the current timestamp, which is a kind of user behavior prediction problem. In this task, a user may interact with an item for many times.

The results are shown in Table 3(b), in which the results of baselines are also reported by CoPE [37]. There are no user/item attributes in Wikipedia and LastFm, so we use all proposed modules except the attribute-integrated SVD in FreeGEM. Since JODIE, CoPE and FreeGEM can update models in test time, they significantly outperform the other methods without test time training. FreeGEM can outperform JODIE and achieve comparable accuracy with CoPE. Although the accuracy of FreeGEM has no obvious advantage over CoPE in this task, we will show later that FreeGEM can significantly outperform CoPE in computation efficiency due to no learnable parameters.

## 5.3 Ablation Studies

**Future item recommendation** We use A, B, C to refer to ablative variants of FreeGEM, and check or cross marks indicate whether the corresponding module exists. As shown in Table 4, when the personalized interaction pattern modeller or frequency-aware reconstruction module is not adopted, the results are suboptimal, which confirms the effectiveness of these two modules.

**Next interaction prediction** We use D-I to refer to ablative variants of FreeGEM, and check or cross marks indicate whether the corresponding module exists. In addition, we use an intuitively effective baseline, called *Last-k*, which takes the recent $k$ items that users interacted with as predictions. The

Table 5: The ablation studies on the next interaction prediction task.

| | Offline | Online | Decay | Attention | Wikipedia MRR | Wikipedia Hit@10 | LastFM MRR | LastFM Hit@10 |
|---|---|---|---|---|---|---|---|---|
| Last-1 | - | - | - | - | 0.775* | 0.775* | 0.098* | 0.098* |
| Last-10 | - | - | - | - | 0.792 | 0.842 | 0.139 | 0.263 |
| D | ✗ | ✗ | ✗ | ✗ | 0.441 | 0.620 | 0.074 | 0.186 |
| E | ✓ | ✗ | ✗ | ✗ | 0.510 | 0.703 | 0.089 | 0.222 |
| F | ✗ | ✓ | ✗ | ✗ | 0.497 | 0.715 | 0.104 | 0.251 |
| G | ✓ | ✓ | ✗ | ✗ | 0.530 | 0.739 | 0.11 | 0.256 |
| H | ✓ | ✓ | ✓ | ✗ | 0.779 | 0.851 | 0.163 | 0.348 |
| I | ✓ | ✓ | ✗ | ✓ | 0.541 | 0.747 | 0.190 | 0.446 |
| **FreeGEM** | ✓ | ✓ | ✓ | ✓ | 0.786 | 0.852 | 0.195 | 0.453 |

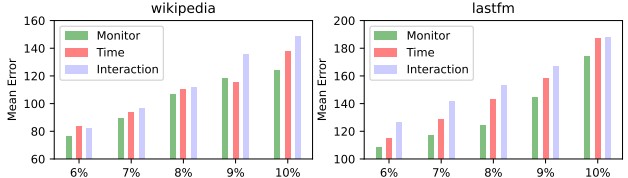
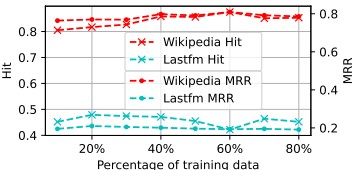

Figure 6: Performance comparison of three restart methods.   Figure 7: Robustness.

results are shown in Table 5. Note that the result of *Last-1* is Hit@1, so we marked its result with *. D dose not update in test phase; E only uses offline SVD to update guided by Monitor in test phase; F updates in real time, but without offline restart. G is better than D, E and F, which confirms the effectiveness of the Online-Monitor-Offline architecture. FreeGEM is better than H and I, indicating the effectiveness of both dynamic time decay module and attention module. Dynamic time decay is very effective on Wikipedia, while attention has stronger effects on LastFm. This is due to the phenomenon that many users continuously interact with the same item in Wikipedia, which can be proved by the excellent results of *Last-k* on Wikipedia. It should be emphasized that, we think the upstream incremental graph embedding engine and the downstream personalized dynamic interaction pattern modeller are independent, and different implementations of the downstream modules can make our model deal with different downstream tasks. Therefore, during the ablation study, we do not perform cross-component ablation studies between the upstream engine and the downstream modeller, but only perform ablation studies within each component (i.e., within the upstream engine and within the downstream modeller).

**Monitor**  We compare the average approximation errors of Monitor, *Interaction* (restart with fixed number of interactions) and *Time* (restart with fixed time), when the number of restarts are the same. First, we execute online SVD to obtain the total approximation error, take 6% - 10% of the total error as the threshold of Monitor, and finally run Time and Interaction according to the number of restarts of Monitor. The results are shown in Figure 6. Compared with the average approximation errors of Monitor, the errors of Interaction and Time are 10.93% and 5.28% higher on Wikipedia and 16.80% and 9.67% higher on LastFm, respectively, indicating that Monitor is more effective.

## 5.4 Other Studies

Table 6: Total running time comparison. The times of speedup is shown in the parentheses.

| Wikipedia | | | Lastfm | | |
|---|---|---|---|---|---|
| FreeGEM | JODIE | CoPE | FreeGEM | JODIE | CoPE |
| 9.7min | 350.0min (36.1X) | 3,589.1min (370.0X) | 54.8min | 15,790.0min (288.1X) | 51,212.5min (934.5X) |

**Running time**  We use the next interaction prediction task to study the efficiency of JODIE, CoPE and FreeGEM. Other baselines, such as Time-LSTM [40], RRN [32], LatentCross [1] and CTDNE [17] show comparable efficiency with JODIE [14] and thus are omitted. JODIE and CoPE both run 50 epochs and choose the best performing model on validation set as the optimal model, but FreeGEM only requires multiple runs of offline/online SVD. As shown in Table 6, FreeGEM is at least 36X faster than JODIE and at least 370X faster than CoPE, demonstrating its high efficiency. Besides, the running time of JODIE and CoPE increase significantly when the number of interactions

increases, but the running time of FreeGEM only increases moderately which is more desirable for real-world applications.

**Cold-start**   We use the future item recommendation task to study the effects of attribute-integrated SVD in cold-start setting. We define the users who have not appeared in training set and verification set as cold-start users. For ML-1M, there are 29 cold-start users. When attributes are not integrated, FreeGEM randomly recommends items to these users. For 17 of the cold-start users whose Recall@10 is not 0, FreeGEM increases their average Recall@10 from 0.011 to 0.029, achieving a relative increase of **166%**. For 7 cold-start users whose Recall@10 is 0, FreeGEM increases their average Recall@10 to **0.100**, which is very high compared with the average Recall@10 of all users in ML-1M.

**Robustness**   For the next interaction prediction task, we change the proportion of the training set to verify the robustness of FreeGEM in different levels of data sparsity. We change the percentage of the training set from 10% to 80%, next 10% interactions after the training set as the validation set, and next 10% interactions after the validation set as the test set. The results are shown in Figure 7, in which we can observe that the accuracy of FreeGEM is almost unaffected. This experiment demonstrate that that FreeGEM has strong robustness to the scale of training data. The results of many baselines can also be found in JODIE [14].

**Under represented groups**   There could be recommendation bias concerns on under represented groups in specific applications and the existence of under represented groups usually comes from the biases during the data collection process. Some bias mitigation techniques are specially designed to solve this problem [21]. We carry out further experiments on ML-100K. First, we group the users by gender and find that there are 670 male users (71.0%) and 273 female users (29.0%) in this dataset. Male users had 753,313 interaction records (74.4%), and female users had 246,298 interaction records (25.6%). This shows that there is bias of gender distribution in this dataset. Then, we remove the attribute-integrated SVD module from our method and calculate the performance of our model on male and female users. We find that without using attributes, the Recall of male users is 0.124 and Recall of female users is 0.085. Male users are with 45.9% higher Recall than female users. Finally, we take the attribute-integrated SVD module back to see how the performance differs. We find that after using attributes, the Recall of male users increases to 0.156, Recall of female users increases to 0.127. Male users are with 22.8% higher Recall than female users. To sum up, due to the bias of gender distribution in the ML-100K dataset, there is a clear bias in recommendation accuracy, i.e., male users are with 45.9% higher Recall than female users even without using user attributes in the model. However, to our surprise, the bias in the recommendation results becomes less significant after using user attributes in our model, i.e., male users are with only 22.8% higher Recall than female users when user attributes are incorporated. We think the reason is that user attributes can help these under represented groups (female users) to improve their embedding quality.

## 6   Conclusion

We propose FreeGEM, a parameter-free dynamic graph embedding method for link prediction. By modelling collaborative relationships and personalized dynamic interaction patterns separately, FreeGEM can alleviate the noises when the two key factors diverge, leading to higher accuracy. Specifically, we propose an incremental graph embedding engine for real-time dynamic graph embedding via a novel Online-Monitor-Offline architecture and a personalized dynamic interaction pattern modeller with dynamic time decay and attention. Since there are no learnable parameters, FreeGEM can avoid the time-consuming back-propagation, leading to significantly higher efficiency. One limitation of this work is that we only explore the link prediction task, leaving other downstream machine learning tasks, such as node classification, as future work. Another line of future work is to extend the high level design of our method to other kinds of parameterized models. Although we empirically find that our method can mitigate rating bias by utilizing user/item attributes, practitioners should pay more attention to the biases during data collection before using our method.

### Acknowledgments and Disclosure of Funding

This work was supported by the National Natural Science Foundation of China (NSFC) under Grants 61932007 and 62172106.

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
