# A  Discussion

## A.1  Connection between SVD and Frequency Analysis

First, we introduce the concept of "low frequency signals" following the common practice in graph signal processing.

We use $R$ to represent the observed interaction matrix, $\tilde{R}$ to represent the user's real preference matrix, where larger $\tilde{R}_{ij}$ indicates higher chance for user $i$ to interact with item $j$, and $\hat{R}$ to represent the predicted interaction matrix. We can use $S = R^\top R$ to represent the similarity between items, and $D$ to represent the degree matrix of $S$. The Laplace matrix of $S$ is defined as $L = D - S$. We can take $\boldsymbol{x} \in \mathbb{R}^n$ as a graph signal where each node is assigned with a scalar. The smoothness of the graph signal can be measured by the total variation defined as follows:

$$TV(\boldsymbol{x}) = \boldsymbol{x}^T L \boldsymbol{x} = \sum S_{ij}(x_i - x_j)^2. \tag{9}$$

When the input graph signal is the real preference vector of user $u$, which means $\boldsymbol{x} = \tilde{R}_u \in \mathbb{R}^n$: 1) if two items $i$ and $j$ $(i \neq j)$ are similar, i.e., $S_{ij}$ is large, the user's preferences for these two items will be similar, which means that $(\tilde{R}_{ui} - \tilde{R}_{uj})^2$ should be small and will not lead to excessive $TV(\boldsymbol{x})$; 2) if two items $i$ and $j$ $(i \neq j)$ are dissimilar, i.e., $S_{ij}$ is small, the user often has different preferences for the two items, which means that $(\tilde{R}_{ui} - \tilde{R}_{uj})^2$ should be large, which also does not cause $TV(\boldsymbol{x})$ to be too large because their similarity $S_{ij}$ is small. In conclusion, if the real preference signal is used as input, the total variation should have a small value. However, due to the exposure noise and quantization noise in the observed interaction matrix [35], the total variation becomes larger when the input signal is the observed user interaction signal, which means $\boldsymbol{x} = R_u$. Therefore, the key to predict the real preference matrix through the observed interaction matrix is to design a low-pass filter to remove the high-frequency part of the observed interaction matrix.

Then, we explain why the reconstruction matrix obtained by truncated SVD is low-frequency, which is also related to graph signal processing.

The energy of the graph signal is defined as $E(\boldsymbol{x}) = ||\boldsymbol{x}||^2$. The normalized total variation of $\boldsymbol{x}$ can be calculated with the Rayleigh quotient as

$$Ray(\boldsymbol{x}) = \frac{TV(\boldsymbol{x})}{E(\boldsymbol{x})} = \frac{\boldsymbol{x}^T L \boldsymbol{x}}{\boldsymbol{x}^T \boldsymbol{x}} = \frac{\sum S_{ij}(x_i - x_j)^2}{\sum x_i^2}. \tag{10}$$

As $L$ is real and symmetric, its eigendecomposition is given by $L = U \Lambda U^T$ where $\Lambda = diag(\lambda_1, \lambda_2, ..., \lambda_n), \lambda_1 \leq \lambda_2 \leq ... \leq \lambda_n$, and $U = (\boldsymbol{u}_1, \boldsymbol{u}_2, ..., \boldsymbol{u}_n)$ with $\boldsymbol{u}_i \in \mathbb{R}^n$ being the eigenvector for eigenvalue $\lambda_i$. We call $\tilde{\boldsymbol{x}} = U^T \boldsymbol{x}$ as the graph Fourier transform of the graph signal $\boldsymbol{x}$ and its inverse transform is given by $\boldsymbol{x} = U \tilde{\boldsymbol{x}}$. Rayleigh quotient can be transformed into spectral domain as

$$Ray(\boldsymbol{x}) = \frac{\boldsymbol{x}^T L \boldsymbol{x}}{\boldsymbol{x}^T \boldsymbol{x}} = \frac{\boldsymbol{x}^T U \Lambda U^T \boldsymbol{x}}{\boldsymbol{x}^T U U^T \boldsymbol{x}} = \frac{\tilde{\boldsymbol{x}}^T \Lambda \tilde{\boldsymbol{x}}}{\tilde{\boldsymbol{x}}^T \tilde{\boldsymbol{x}}} = \frac{\sum \lambda_i \tilde{x}_i^2}{\sum \tilde{x}_i^2}. \tag{11}$$

Take $\boldsymbol{x} = \boldsymbol{u}_i$, we can get $Ray(\boldsymbol{u}_i) = \lambda_i$, indicating that the eigenvector corresponding to the small eigenvalue is smoother.

There are similar conclusions when we take $S = RR^\top$. SVD extends the signal on the node from scalar to vector. The three matrices obtained by truncated SVD correspond to the first $k$ eigenvectors of $RR^\top$, the first $k$ eigenvalues of $RR^\top$ and the first $k$ eigenvectors of $R^\top R$ respectively. Therefore, it only retains the eigenvectors with low frequency to reconstruct the interaction matrix, so its essence is an ideal low-pass filter. And their frequency is related to the magnitude of eigenvalues.

Recently, Nt et al. [18] show that the method of graph neural network is essentially a low-pass graph filter. Shen et al. [23] show that matrix factorization methods, linear auto-encoder methods, and neighborhood-based methods can be equivalently described by designing different forms of graph filters in graph signal processing. In addition, the method based on matrix factorization is proved to be equivalent to an infinite layer graph neural network.

## A.2 Why $\pi/3$ is the Boundary?

In summary, $\pi/3$ is the threshold to determine the effectiveness of model updates. Online updating towards angles greater than $\pi/3$ indicates the online updating is even worse than no updating at all. More detailed discussion is presented below.

When we use offline SVD during the processes of $\hat{R}_0$->$\hat{R}_1$, $\hat{R}_1$->$\hat{R}_2$ and $\hat{R}2$->$\hat{R}_3$, there will be no approximation error. However, the online SVD used in the processes of $\hat{R}_0$->$\hat{R}_{1'}$, $\hat{R}_{1'}$->$\hat{R}_{2'}$ and $\hat{R}_{2'}$->$\hat{R}_{3'}$ has approximation errors. Thus, their evolution directions are not consistent. We use the F-norm of the difference between $\hat{R}_i$ and $\hat{R}_{i'}$ ($i = 1, 2, 3$) to measure the online approximation error. We cannot directly calculate this value in most cases, because the model will only execute the Online module in most cases. From Figure 4, we find that the F-norm of the difference between $\hat{R}_0$ and $\hat{R}_{i'}$ (defined as *distance* in this paper) is positively correlated with the F-norm of the difference between $\hat{R}_i$ and $\hat{R}_{i'}$ ($i = 1, 2, 3$), and we have verified this empirically in Appendix A.3. Thus, we use *distance* to estimate the *error*. As the approximation of the Offline module, the approximated value calculated by the Online module should not be worse than the result without updating, otherwise it means that the Online module is invalid. In this case, the approximation error of the reconstructed matrix ($\hat{R}_{1'}$, $\hat{R}_{2'}$, $\hat{R}_{3'}$) obtained by the Online module is even greater than the F-norm between the original matrix ($\hat{R}_0$) and the reconstructed matrix ($\hat{R}_1$, $\hat{R}_2$, $\hat{R}_3$) obtained by the Offline module. Figure 8 shows the case when the evolution direction of Offline module and Online module is greater than $\pi/3$. It can be seen that the F-norm of the difference between $\hat{R}_0$ and $\hat{R}_i$ is smaller than the F-norm of the difference between $\hat{R}_i$ and $\hat{R}_{i'}$ ($i = 1, 2, 3$). In this case, instead of using $\hat{R}_{i'}$ as the approximation of $\hat{R}_i$, it is better to directly use $\hat{R}_0$ as its approximation, i.e., online updating is even worse than no updating at all.

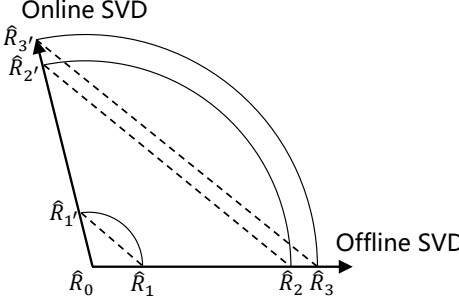

Figure 8: The evolution direction of Offline module and Online module is greater than $\pi/3$

## A.3 The Positive Correlation Between Approximation Error and Distance

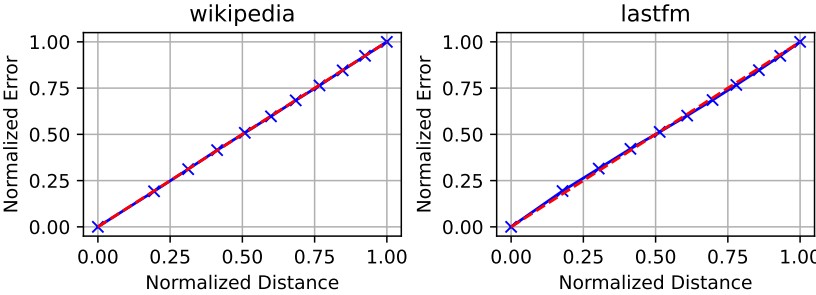

Figure 9: There is a strong positive correlation between *distance* (the F-norm distance between the reconstructed matrix by online SVD at the current time step and the initial reconstructed matrix) and online approximation error.

We first divide each dataset into $T = 100$ time intervals according to the number of interactions, and run offline SVD as ground truth at the end of each time interval. We start to execute the online SVD at the end of $t$-th time interval, and calculate the error of the online SVD and the *distance* at the end of $t + \Delta$-th time interval. The value of $\Delta$ is from 1 to 10, and the corresponding value of $t$ is from 1 to $T - \Delta$. Each $\Delta$ corresponds to a group of errors and a group of *distance*. By grouping according to $\Delta$, we can calculate the mean value of each group of errors and the mean value of *distance*, respectively, and draw the curve to understand their relationship. For $\Delta = 1, ..., 10$, we define $d_\Delta$ and $e_\Delta$ as the corresponding mean of *distance* and mean of online approximation errors. Then, we normalize $d_\Delta$ and $e_\Delta$ as the $x$-axis and $y$-axis, respectively, as follows:

$$x_\Delta = ||d_\Delta|| = \frac{d_\Delta^m}{max(d_1^m, ..., d_\Delta^m)}, \tag{12}$$

$$y_\Delta = ||e_\Delta|| = \frac{e_\Delta}{max(e_1, ..., e_\Delta)}. \tag{13}$$

As shown in Figure 9, there is a positive correlation between the normalized error and the normalized *distance*, indicating that it is reasonable to estimate the online approximation error using the *distance* measure. Although the power $m$ varies between the two datasets, we can always find an appropriate $m$ so that we can fit the relationship between $d_\Delta$ and $e_\Delta$ almost by a straight line.

### A.4 Memoryless Property of Time Decay Function

The Online-Monitor-Offline architecture indicates that our model has the concept of stage. We believe that the decay function should have no memories, that is, the decay ratio of the same time interval should be treated consistently in different stages. The reasons are explained in the following discussion.

Suppose there are four timestamps $t_1$, $t_2$, $t_3$, $t_4$, and $t_2 - t_1 = t_4 - t_3$. The decay function we used $f_i(t) = exp\{\beta_i(t/T_i - 1)\}$ is memoryless. When these four timestamps are all in the same stage (e.g., $i$-th stage), we have $f_i(t_2)/f_i(t_1) = f_i(t_4)/f_i(t_3)$ using the decay function. When these four timestamps are in different stages, (e.g., $t_1$ and $t_2$ are at i-th stage, and $t_3$ and $t_4$ are at j-th stage ($i \neq j$)), we have $f_i(t_2)/f_i(t_1) = f_j(t_4)/f_j(t_3)$, when $\beta_i/\beta_j = T_i/T_j$.

We point out here that the linear decay function $g_i(t) = \beta_i t/T_i$ is not memoryless. This is because we cannot ensure $g_i(t_2)/g_i(t_1) = g_j(t_4)/g_j(t_3)$ both in the above two cases.

### A.5 Comparison between Different Models

In order to more clearly explain the difference between FreeGEM and TPP-based, RNN-based and GNN-based methods, we summarized this and presented the results in the Table 7.

In all methods, only FreeGEM is parameter-free. Various dynamic graph learning methods use time decay mechanism, but in different forms. For the methods based on temporal point process [28, 30, 41], the time decay is reflected in the intensity function. When RNN-based method [14] predicts the user/item embeddings, it specifically considers the current user's previous interactions using RNN. GNN-based method [37] adopts a neural ordinary differential equation to model the temporal dynamics of user/item embeddings. It should be noted that the dynamic time decay in other methods can also be applied to our framework but may introduce learnable-parameters and thus hurt the computational efficiency.

TPP-based models [28, 30, 41] make use of collaboration through interaction between users and items. RNN-based model [14] uses a pair of coupled RNNs to model users and items respectively to make use of collaborative relationships. Because GNN naturally contains graph information, the model based on GNN [37] naturally uses the collaborative relationship through adjacency matrix. Similar to the GNN-based model, the SVD method used by FreeGEM also contains graph information, so collaborative information is naturally adopted.

Table 7: Comparison between different kinds of methods.

| | TPP-based | RNN-based | GNN-based | FreeGEM |
|---|---|---|---|---|
| parameter-free | ✗ | ✗ | ✗ | ✓ |
| how to use time information | intensity function | RNN | neural ordinary differential equation | dynamic time decay |
| how to use collaboration relationship | interaction events | coupled RNN | GNN | SVD |

# B Appendix

## B.1 Statistics of the Datasets

We use four publicly available datasets to evaluate the performance of FreeGEM in the future item recommendation task, the detailed statistics of which are presented in Table 8.

Table 8: Statistics of the datasets of the future item recommendation task.

| Datasets | # Users | # Items | # Interactions | # Density | # Unique Timestamps |
|---|---|---|---|---|---|
| Amazon Video | 5,130 | 1,685 | 37,126 | 0.43% | 1,946 |
| Amazon Game | 24,303 | 10,672 | 231,780 | 0.09% | 5,302 |
| MovieLens-100K | 943 | 1,349 | 99,287 | 7.81% | 49,119 |
| MovieLens-1M | 6,040 | 3,416 | 999,611 | 4.85% | 458,254 |

We use two publicly available datasets to evaluate the performance of FreeGEM in the next interaction prediction task, the detailed statistics of which are presented in Table 9.

Table 9: Statistics of the datasets of the next interaction prediction task.

| Dataset | # Users | # Items | # Interactions | # Unique Timestamps |
|---|---|---|---|---|
| Wikipedia | 8,227 | 1,000 | 157,474 | 152,757 |
| LastFM | 980 | 1,000 | 1,293,103 | 1,283,614 |

## B.2 Hyperparameter Settings

Although there are no learnable parameter in FreeGEM, we have several hyperparameters for model building and user preference fusion. In our experiments, we use a simple grid search method to obtain the optimal hyperparameters. The hyperparameter search space of FreeGEM in the future item recommendation task is presented in Table 10.

Table 10: Hyperparameter search space for future item recommendation task.

| $\beta_1$ | $\alpha$ | $k_1$ | $k_2, k_3, k_4, k_5$ | $\alpha_1$ | $\alpha_2$ | $\alpha_3$ |
|---|---|---|---|---|---|---|
| 1, ..., 100 | 2.0 | 1, 2, 4, 8, 16, 32, 64, 128, 256 | 0, 1 | 0, 1, 2 | 0, 1, 2 | 0, 1, 2 |

After grid search, we find the optimal hyperparameters of FreeGEM for the four datasets on the future item recommendation task in Table 11.

Table 11: Hyperparameter settings for future item recommendation task.

| | $\beta_1$ | $\alpha$ | $k_1$ | $k_2$ | $k_3$ | $k_4$ | $k_5$ | $\alpha_1$ | $\alpha_2$ | $\alpha_3$ |
|---|---|---|---|---|---|---|---|---|---|---|
| Video | 21.0 | 2.0 | 128 | 0 | 0 | 0 | 0 | 1 | 0 | 0 |
| Game | 18.0 | 2.0 | 256 | 0 | 0 | 0 | 0 | 1 | 0 | 0 |
| ML-1M (no-attr) | 60.0 | 2.0 | 8 | 0 | 0 | 0 | 0 | 1 | 0 | 0 |
| ML-100K (no-attr) | 60.0 | 2.0 | 1 | 0 | 0 | 0 | 0 | 1 | 0 | 0 |
| ML-1M (with-attr) | 50.0 | 2.0 | 4 | 1 | 1 | 1 | 1 | 0 | 1 | 0 |
| ML-100K (with-attr) | 15.0 | 2.0 | 1 | 1 | 1 | 1 | 1 | 0 | 2 | 1 |

The hyperparameter search space of FreeGEM in the next interaction prediction task is presented in Table 12.

Table 12: Hyperparameter search space for next interaction prediction task.

| | $d$ | $\beta_1$ | $a$ | $b$ | $k_1$ | $\gamma$ | $\lambda$ |
|---|---|---|---|---|---|---|---|
| Wikipedia | 5, 10,..., 50 | 5, 10, ..., 50 | 1, 3, 5 | 1, 2, 3 | 128, 256, 512 | 1/2, 1/3, ..., 1/10 | 0.01, 0.02, ..., 1.00 |
| Lastfm | 300, 400, ..., 800 | 1, 2, ..., 10 | 1, 3, 5 | 1, 2, 3 | 128, 256, 512 | 1/2, 1/3, ..., 1/10 | 0.01, 0.02, ..., 1.00 |

After grid search, we find the optimal hyperparameters of FreeGEM for the two datasets on the next interaction prediction task in Table 13.

Table 13: Hyperparameter settings for next interaction prediction task.

| | $d$ | $\beta_1$ | $a$ | $b$ | $k_1$ | $\gamma$ | $\lambda$ |
|---|---|---|---|---|---|---|---|
| Wikipedia | 35.0 | 35.0 | 3 | 1 | 512 | 1/2 | 0.80 |
| LastFM | 500.0 | 2.0 | 1 | 2 | 512 | 1/5 | 0.74 |

In addition, we have the following observations about the hyperparameters.

(1) Hyperparameters related to the attribute-integrated SVD module include $k_2, k_3, k_4, k_5$, and $\alpha_1, \alpha_2, \alpha_3$. Due to the inclusion of attribute information, this module can improve the prediction accuracy but also introduce several more hyperparameters. It can be observed in ablation experiments that our method can still surpass other methods even without using attribute information. Thus, in practice, attribute information can be used only for those users with scarce interactions or cold start users, which can significantly improve the accuracy as shown in Section 5.4. In addition, during hyperparameter search, we observe that using $k_1 = k_2 = k_3 = k_4$ can achieve outstanding performance, even though they can be tuned separately.

(2) The hyperparameter that controls the restart of the Offline module has less significant effect on the prediction accuracy. Its role is to control the restart times according to the data scale. As we can see, in Wikipedia dataset with small data scale, we use the search interval of 5, 10, ..., 50, while in LastFm dataset with large data scale, we use the search interval of 300, 400, ..., 800.

(3) In the experiments, we find that for different datasets, the hyperparameter $\beta$, which controls the time decay is sensitive to the dataset. We can set higher priority for the searching of $\beta$. Luckily, $\beta$ is not very sensitive to the values of other hyperparameters, so that we can search other hyperparameters after the optimal $\beta$ is found.

(4) As shown in Table 6, the model training time of our method is much shorter compared to other methods for one group of hyperparameters, especially on larger dataset. Thus, we find that the overall running time (include hyperparameter searching) of our method is still much lower than the other methods.

### B.3 Experimental Environment

We run all the experiments on a server equipped with one NVIDIA TESLA T4 GPU and Intel(R) Xeon(R) Gold 5218R CPU. All the code of this work is implemented with Python 3.9.7.

### B.4 Copyrights of the Existing Assets

All the code that we use to reproduce the results of the compared works is publicly available and permits usage for research purpose.

All the datasets that we use in the experiments are publicly available and permit usage for research purpose.