# OpenReview forum: "Parameter-free Dynamic Graph Embedding for Link Prediction"
_NeurIPS.cc/2022/Conference — NeurIPS 2022 Accept_

### Official Review · Reviewer_vE5r · 2022-07-07

**Rating:** 7
**Confidence:** 3
**Soundness:** 3 good
**Presentation:** 3 good
**Contribution:** 3 good

**Summary:**

The authors present a parameter-free graph embedding model to handle the missing link prediction tasks, including the future-item recommendation and the next-item recommendation. In short, the framework consists of 1) a graph embedding module and 2) a personalization module, where the former is built with an online-monitor-offline structure, and the latter introduces the concept of time decay and attention.

**Questions:**

Is it possible to apply such an architecture to parameterized models?


**Limitations:**

Yes.

**Strengths And Weaknesses:**

Strength
- The idea is novel and interesting.
- The proposed model achieves a good improvement in efficiency.
- The experiments are comprehensive, including the evaluation of two link prediction tasks, the ablation studies, and results regarding the efficiency.

Weakness
- Some parts of the paper are unclear.
1. In line 181, why π/3 is the boundary?
2. In line 183, what does 1-3 or 1’-3’ mean?
3. In line 215, there are variant ways to do time decay. Why this one?
4. In table 5, is it possible to apply H and I without D/E/F/G?
- It is a bit hard to totally understand the key to the solution as the whole model contains many submodules, which are not comparable to other existing models. For example, the decay mechanism is essential, but do some other models not consider it? I suggest the authors compile a table explaining the differences among the compared models.

---

> ### Author Response · Authors · 2022-08-02
> **Response to Reviewer vE5r (1/3)**
>
> **Q1:** In line 183, what does 1-3 or 1'-3' mean?
>
> **A1:** "$1$-$3$" corresponds to the three reconstructed matrices obtained by the Offline module after observing three user-item interactions, that is, the ground truth of the prediction matrix. For example, $1$ stands for $\hat{R}_{1}$.
>
> "$1'$-$3'$" corresponds to the three reconstructed matrices obtained by the model using the Online module after observing three user-item interactions, that is, the online estimation of the prediction matrix. For example, $1'$ stands for $\hat{R}_{1'}$.
>
> Maybe using $\hat{R}\_{i}$ ($\hat{R}\_{i'}$) rather than $i$ ($i'$) will make this more clear. We will revise this in the final version.
>
> **Q2:** in line 181, why π/3 is the boundary?
>
> **A2:** In summary, π/3 is the threshold to determine the effectiveness of model updates. Online updating towards angles greater than π/3 indicates the online updating is even worse than no updating at all. More detailed discussion is presented below.
>
> When we use offline SVD during the processes of $0$->$1$, $1$->$2$ and $2$->$3$, there will be no approximation error. However, the online SVD used in the processes of $0$->$1'$, $1'$->$2'$ and $2'$->$3'$ has approximation errors. Thus, their evolution directions are not consistent. We use the F-norm of the difference between $i$ and $i'$ $(i=1,2,3)$ to measure the online approximation error. We cannot directly calculate this value in most cases, because the model will only execute the Online module in most cases. From Figure 4, we find that the F-norm of the difference between 0 and $i'$ (defined as *distance* in this paper) is positively correlated with the F-norm of the difference between $i$ and $i'$ $(i=1,2,3)$, and we have verified this empirically in Appendix A. Thus, we use *distance* to estimate the *error*. As the approximation of the Offline module, the approximated value calculated by the Online module should not be worse than the result without updating, otherwise it means that the Online module is invalid. In this case, the approximation error of the reconstructed matrix $(1', 2', 3')$ obtained by the Online module is even greater than the F-norm between the original matrix ($0$) and the reconstructed matrix $(1,2,3)$ obtained by the Offline module. This figure shows the case when the evolution direction of Offline module and Online module is greater than π/3: https://anonymous.4open.science/r/NeurIPS2022-Paper5388-FreeGEM/monitor.png.
> It can be seen that the F-norm of the difference between $0$ and $i$ is smaller than the F-norm of the difference between $i$ and $i'$ $(i=1,2,3)$. In this case, instead of using $i'$ as the approximation of $i$, it is better to directly use $0$ as its approximation, i.e., online updating is even worse than no updating at all.
>
> **Q3:** In line 215, there are variant ways to do time decay. Why this one?
>
> **A3:** The Online-Monitor-Offline architecture makes our model has the concept of *stage*, and our main contribution is to propose a "dynamic" time decay to fit this architecture. We believe that the decay function should have no memories, that is, the decay ratio of the same time interval should be treated consistently in different stages. The reasons are explained in the following discussion.
>
> Suppose there are four timestamps $t1$, $t2$, $t3$, $t4$, and $t2-t1=t4-t3$.
>
> (1) The decay function we used $f\_i(t)=\exp\{\beta_i(t/T_i -1)\}$ is memoryless. When these four timestamps are all in the same stage (e.g., $i$-th stage), we have $f_i(t2)/f_i(t1)=f_i(t4)/f_i(t3)$ using the decay function. When these four timestamps are in different stages, (e.g., $t1$ and $t2$ are at i-th stage, and $t3$ and $t4$ are at j-th stage ($i\neq j$)), we have $f_i(t2)/f_i(t1)=f_j(t4)/f_j(t3)$, when $\beta_i/\beta_j=T_i/T_j$.
>
> (2) We point out here that the linear decay function $g_i(t)=\beta_it/T_i$ is not memoryless. This is because we cannot ensure $g_i(t2)/g_i(t1)=g_j(t4)/g_j(t3)$ both in the above two cases.
>
> In fact, for any time decay function, it is worth trying as long as it is memoryless and can reduce the effects of long-term interactions and promote the recent interactions. We will discuss this in more details in the final version. Designing more effective decay functions will definitely be one of the most important future works for us.

---

> > ### Author Response · Authors · 2022-08-02
> > **Response to Reviewer vE5r (2/3)**
> >
> > **Q4:** It is a bit hard to totally understand the key to the solution as the whole model contains many submodules, which are not comparable to other existing models. For example, the decay mechanism is essential, but do some other models not consider it? I suggest the authors compile a table explaining the differences among the compared models.
> >
> > **A4:** Thanks for your suggestion. In fact, various dynamic graph learning methods use time decay mechanism, but in different forms from ours. For the methods based on temporal point process [1,2,3], the time decay is reflected in the intensity function. When JODIE [4] predicts the user/item embeddings, it specifically considers the current user's previous interactions using RNN. CoPE [5] adopts a neural ordinary differential equation to model the temporal dynamics of user/item embeddings. It should be noted that the dynamic time decay in other methods can also be applied to our framework but may introduce learnable-parameters and thus hurt the computational efficiency.
> >
> > Following the reviewer's suggestion, we will add a table to compare our method with other related works in the final version.
> >
> > [1] Trivedi, Rakshit, et al. "Know-evolve: Deep temporal reasoning for dynamic knowledge graphs." *international conference on machine learning*. PMLR, 2017.
> >
> > [2] Wang, Yichen, et al. "Coevolutionary latent feature processes for continuous-time user-item interactions." *Advances in neural information processing systems* 29 (2016).
> >
> > [3] Zuo, Yuan, et al. "Embedding temporal network via neighborhood formation." *Proceedings of the 24th ACM SIGKDD international conference on knowledge discovery & data mining*. 2018.
> >
> > [4] Kumar, Srijan, Xikun Zhang, and Jure Leskovec. "Predicting dynamic embedding trajectory in temporal interaction networks." *Proceedings of the 25th ACM SIGKDD international conference on knowledge discovery & data mining*. 2019.
> >
> > [5] Zhang, Yao, et al. "CoPE: Modeling Continuous Propagation and Evolution on Interaction Graph." *Proceedings of the 30th ACM International Conference on Information & Knowledge Management*. 2021.
> >
> > **Q5:** Is it possible to apply such an architecture to parameterized models?"
> >
> > **A5:** We think so. This will also be one of the future works for us. More detailed discussions are as follows.
> >
> > (1) Figure 1 shows the high-level design of our method. The upstream engine provides users and item representations, in which similar items/users have similar embeddings. The downstream modeller directly uses these embeddings to complete the link prediction task. Actually,  the downstream modeller can do different tasks by replacing it with corresponding modules. For example, leveraging a module with classification ability can solve the problem of dynamic graph node classification. The node classification often needs to transform the features of nodes by classifiers, so it is often a parametric model. We think this will be a promising future research direction.
> >
> > It should be noted that this architecture is more suitable for non-parametric models or parameter-efficient models. Because the incremental graph embedding engine gets embedding through SVD calculation, the embedding of users and items naturally lies in the same representation space. Because of this, there is no need to introduce parameter matrix for feature transformation in the downstream modeller.
> >
> > (2) We believe that the Online-Monitor-Offline architecture can be extended to any scenario that meets such conditions (whether it is a parametric model or not): 1) the original algorithm is time-consuming, and similar to the Offline module, it can be sparsely executed; 2) the approximation algorithm is efficient but has cumulative errors, and similar to the Online module, it can be frequently executed. In such a scenario, the design of Monitor can also be leveraged, and the error of Online module can be monitored efficiently following the same idea of our method.

---

> > > ### Author Response · Authors · 2022-08-02
> > > **Response to Reviewer vE5r (3/3)**
> > >
> > > **Q6:** In table 5, is it possible to apply H and I without D/E/F/G?
> > >
> > > **A6:** In Table 5, D/E/F/G/H/I corresponds to six different ablation models. D/E/F/G is used to verify the effectiveness of Online-Monitor-Offline architecture in incremental graph embedding engine, where G also verifies the effectiveness of the downstream modeller. H and I are used to verify the effectiveness of dynamic time decay and attention mechanism in the modeller.
> > >
> > > If we understand correctly, you are asking why we didn't ablate the Online-Monitor-Offline architecture and the modeller at the same time to verify the effectiveness of dynamic time decay and attention mechanism. If this understanding is correct, our answer is: yes, it is possible to apply H and I without D/E/F/G.
> > >
> > > The reason why we didn't do the experiment is that, as mentioned earlier, we think the upstream incremental graph embedding engine and the downstream personalized dynamic interaction pattern modeller are independent, and different implementations of the downstream modules can make our model deal with different downstream tasks. Therefore, during the ablation study, we did not perform cross-component ablation studies between the upstream engine and the downstream modeller, but only performed ablation studies within each component (i.e., within the upstream engine and within the downstream modeller). We can also add the corresponding ablation studies if the reviewer thinks it helpful.

---

### Official Review · Reviewer_coS6 · 2022-07-11

**Rating:** 6
**Confidence:** 4
**Soundness:** 3 good
**Presentation:** 3 good
**Contribution:** 3 good

**Summary:**

This paper introduces a parameter-free dynamic graph embedding (FreeGEM) method for link prediction tasks. For modeling user preference in dynamic integration graphs, there are two crucial factors to consider: 1) collaborative relationships among users and 2) user personalized interaction patterns. Existing methods usually implicitly consider these two factors together and suffer from noisy distributions. The proposed method considers the two factors separately instead: with an incremental graph embedding engine to model the former and a personalized dynamic interaction pattern modeler to model the latter. For the incremental graph embedding engine, an Online-Monitor-Offline architecture is proposed, with an online module to approximate the user/item embeddings, a monitor module to estimate the errors of the online module, and an offline module to calibrate the embeddings, such that the module can learn the embeddings efficiently without exceeding the error threshold. Extensive experiments are conducted to demonstrate the effectiveness and efficiency of the proposed method.

**Questions:**

1) In  section 4.2, it says u^(r) is the user embedding as r-th time step. Since it never appears in the paper before, is u^(r) a learned embedding from the model? In section 3.2, it also uses the notation u as a one-hot vector indicating the user id. Are they the same thing?
2) In section 3.1.1, it is not clear for me how E_I and E_U are involved in obtaining the interaction matrix(R-hat) with the low-frequency approximation. Can you explain more about the logic flow here?
3) Through the description of section 3,  it seems that Online-Monitor-Offline architecture eventually estimates the interaction matrix. How do you transform the interaction matrix to user/item embedding? Some clarifications are recommended here.
4) It would be helpful to label all the equations in this paper.
5) Line 155, a missing space between the sentences.


**Limitations:**

The authors adequately addressed the limitations and potential negative societal impact of their work.

**Strengths And Weaknesses:**

Strengths:

- Originality: The proposed method with the two modules to model the two factors and the Online-Monitor-Offline architecture is novel.

- Quality: The content of the paper is well organized. The motivation is clear. Sufficient experiments and ablation studies are conducted.

- Significance: This paper presents a new method that can estimate the user/item embedding for link prediction tasks efficiently.

Weaknesses:

The methodology part needs more clarification. Some equations or concepts are not well-explained. I point out these issues in Questions.

---

> ### Author Response · Authors · 2022-08-02
> **Response to Reviewer coS6 (1/3)**
>
> **Q1:** Through the description of section 3, it seems that Online-Monitor-Offline architecture eventually estimates the interaction matrix. How do you transform the interaction matrix to user/item embedding? Some clarifications are recommended here.
>
> **A1**: Thanks for your suggestion. We first introduced the classic truncated SVD method in lines 105-108. This method will decompose the interaction matrix to obtain $U$, $S$, $V$ so that $R\approx USV^\top$, and then get user embedding and item embedding by $E_U=US^{1/2}$，$E_ I=VS^{1/2}$.
>
> Second, we propose the frequency-aware preference matrix reconstruction (Section 3.1.1) to alleviate the oversmoothing problem, that is, we use $E_U=US^\gamma$ and $E_I=VS^\gamma$ to obtain user embedding and item embedding, in which $U$, $S$, $V$ are obtained by the previous truncated SVD.
>
> Third, we propose the attribute-integrated SVD (Section 3.1.2) to integrate attribute information into the model. Equation (1) and (2) show how the Offline module obtains user/item embeddings. In this component, not only the interaction matrix but also the attribute information of users and items are adopted. The only difference between Online module and Offline module is the way to obtain $S$, $U$, $V$.
>
> We think the confusion mainly comes from the missing connections between truncated SVD and our method. Due to the space limitation, we omitted the details of the the truncated SVD, but we will introduce more details of truncated SVD and its connections with our method in the appendix of the final version.

---

> > ### Author Response · Authors · 2022-08-02
> > **Response to Reviewer coS6 (2/3)**
> >
> > **Q2:** In section 3.1.1, it is not clear for me how E_I and E_U are involved in obtaining the interaction matrix(R-hat) with the low-frequency approximation.
> >
> > **A2:** First, we introduce the concept of "low frequency signals" following the common practice in graph signal processing.
> >
> > We use $R$ to represent the observed interaction matrix, $\tilde{R}$ to represent the user's real preference matrix, where the larger $\tilde{R}\_{ij}$, the easier it is for user $i$ to interact with item $j$, and $\hat{R}$ to represent the predicted interaction matrix. We can use $S=R^\top R$ to represent the similarity between items, and $D$ to represent the degree matrix of $S$. The Laplace matrix of $S$ is defined as $L=D-S$.  We can take $\pmb{x}\in\mathbb{R}^n$ as a graph signal where each node is assigned with a scalar. The smoothness of the graph signal can be measured by the total variation $$TV(\pmb{x})=\pmb{x}^TL\pmb{x}=\sum S_{ij}(x_i-x_j)^2.$$ When the input graph signal is the real preference vector of user $u$, which means $\pmb{x}=\tilde{R}\_{u}\in\mathbb{R}^n$: 1) if two items $i$ and $j$ $(i\not=j)$ are similar, i.e., $S_{ij}$ is large, the user's preferences for these two items will be similar, which means that $(\tilde{R}\_{ui}-\tilde{R}\_{uj})^2$ should be small and will not lead to excessive $TV(\pmb{x})$; 2) if two items $i$ and $j$ $(i\not=j)$ are dissimilar, i.e., $S\_{ij}$ is small, the user often has different preferences for the two items, which means that $(\tilde{R}\_{ui}-\tilde{R}\_{uj})^2$ should be large, which also does not cause $TV(\pmb{x})$ to be too large because their similarity $S_{ij}$ is small. In conclusion, *if the real preference signal is used as input, the total variation should have a small value*. However, due to the exposure noise and quantization noise in the observed interaction matrix[1], *the total variation becomes larger when the input signal is the observed user interaction signal*, which means $\pmb{x}=R_u$. Therefore, the key to predict the real preference matrix through the observed interaction matrix is to design a low-pass filter to remove the high-frequency part of the observed interaction matrix.
> >
> > Then, we explain why the reconstruction matrix obtained by truncated SVD is low-frequency, which is also related to graph signal processing.
> >
> > The energy of the graph signal is defined as $E(\pmb{x})=||\pmb{x}||^2$. The normalized total variation of $\pmb{x}$ can be calculated with the Rayleigh quotient as $$Ray(\pmb{x})=\frac{TV(\pmb{x})}{E(\pmb{x})}=\frac{\pmb{x}^TL\pmb{x}}{\pmb{x}^T\pmb{x}}=\frac{\sum S_{ij}(x_i-x_j)^2}{\sum x_i^2}.$$
> > As $L$ is real and symmetric, its eigendecomposition is given by $L=U\Lambda U^T$ where $\Lambda=diag(\lambda_1,\lambda_2,...,\lambda_n),\lambda_1\le \lambda_2\le...\le \lambda_n$, and $U=(\pmb{u}_1,\pmb{u}_2,...,\pmb{u}_n)$ with $\pmb{u}_i\in\mathbb{R}^{n}$ being the eigenvector for eigenvalue $\lambda_i$. We call $\tilde{\pmb{x}}=U^T\pmb{x}$ as the graph Fourier transform of the graph signal $\pmb{x}$ and its inverse transform is given by $\pmb{x}=U\tilde{\pmb{x}}$. Rayleigh quotient can be transformed into spectral domain as $$Ray(\pmb{x})=\frac{\pmb{x}^TL\pmb{x}}{\pmb{x}^T\pmb{x}}=\frac{\pmb{x}^TU\Lambda U^T\pmb{x}}{\pmb{x}^TUU^T\pmb{x}}=\frac{\tilde{\pmb{x}}^T\Lambda \tilde{\pmb{x}}}{\tilde{\pmb{x}}^T\tilde{\pmb{x}}}=\frac{\sum\lambda_i\tilde{x}^2_i}{\sum\tilde{x}^2_i}.$$ Take $\pmb{x}=\pmb{u}_i$, we can get $Ray(\pmb{u}_i)=\lambda_i$, indicating that the eigenvector corresponding to the small eigenvalue is smoother.
> >
> > There are similar conclusions when we take $S=RR^\top$. SVD extends the signal on the node from scalar to vector. The three matrices obtained by truncated SVD correspond to the first $k$  eigenvectors of $RR^\top$, the first $k$ eigenvalues of $RR^\top$ and the first $k$ eigenvectors of $R^\top R$ respectively. Therefore, it only retains the eigenvectors with low frequency to reconstruct the interaction matrix, so its essence is an ideal low-pass filter. And their frequency is related to the magnitude of eigenvalues.
> >
> > Recently, Nt et al. [2] show that the method of graph neural network is essentially a low-pass graph filter. Shen et al. [3] show that the method based on matrix factorization, the method based on linear auto-encoder, and neighborhood-based approaches are summarized into different forms of low-pass graph filters. In addition, the method based on matrix factorization is proved to be equivalent to an infinite layer graph neural network.
> >
> > We will add these discussions and references in the final version for better clarification.
> >
> > [1] Yu, Wenhui, and Zheng Qin. "Graph convolutional network for recommendation with low-pass collaborative filters." *ICML*. 2020.
> >
> > [2] Nt, Hoang, and Takanori Maehara. "Revisiting graph neural networks: All we have is low-pass filters." *arXiv preprint arXiv:1905.09550* (2019).
> >
> > [3] Shen, Yifei, et al. "How Powerful is Graph Convolution for Recommendation?." *CIKM*. 2021.

---

> > > ### Author Response · Authors · 2022-08-02
> > > **Response to Reviewer coS6 (3/3)**
> > >
> > > **Q3:** In section 4.2, it says u^(r) is the user embedding as r-th time step. Since it never appears in the paper before, is u^(r) a learned embedding from the model? In section 3.2, it also uses the notation u as a one-hot vector indicating the user id. Are they the same thing?
> > >
> > > **A3:** Sorry for the confusion. They are not the same thing. $u^{(r)}$ in Section 4.2 is the embedding generated by the Online module, corresponding to a row of $E_U$, where $r$ represents the time step from the current time. As mentioned by the reviewer, the notation $\pmb{u}$ in Section 3.2 is the one-hot vector indicating the user id.
> > > We will refine the notations in the final version for better clarification.
> > >
> > > **Q4:** It would be helpful to label all the equations in this paper.
> > >
> > > **A4:** Thanks for your suggestion. We will label all the equations in the final version.
> > >
> > > **Q5:** Line 155, a missing space between the sentences.
> > >
> > > **A5:** Thanks for pointing out this typo. We will fix this as well as all other typos in the final version.

---

> > > > ### Comment · Reviewer_coS6 · 2022-08-08
> > > > **Thank you**
> > > >
> > > > Thank you for the clarification and efforts to address my comments.

---

### Official Review · Reviewer_uRAL · 2022-07-18

**Rating:** 5
**Confidence:** 3
**Soundness:** 2 fair
**Presentation:** 2 fair
**Contribution:** 2 fair

**Summary:**

This work proposed an approach to model the dynamic user-item interaction graphs. Ths proposed approach is a framework comprising an online-offine architechture with majorly matrix-factorization techniques.

**Questions:**

I understand it is tedious to prepare the code and data sets. But I still strongly suggest the authors provide the code and data sets to make the process more transparent. Maybe there are some concerns (e.g. ethical, anonymity) the author(s) can elaborate more in the author rebuttal phase?

**Limitations:**

I have my reservation related to the content of checklist where the author(s) wrote "To the best of our knowledge, this work does not have potential negative social impacts." I like the confidence signaled. However, when it comes to user-item data analysis, a common ethical issue is that some users and items may suffer because they belong to some under represented groups. Especially when the training data is splited into fragments. I think the paper can be better if the author(s) elaborate more on this direction and futhermore the potentials of this work or continuations of this work can help mediate the mentioned issue.

**Strengths And Weaknesses:**

I like the combination of matrix factorization and dynamic graph-structured data. The experiment results demonstrate that this combination does have a positive impact.

The idea of parameter-free is always exciting in data analysis. However, I cannot fully agree with the definition proposed by the author(s). I expect to see a model without or with less complicated hyper-parameter or training parameter set-ups. However, according to the definition:

"...we do not incorporate any parameters which need to be learned via back-propagation..." the reality is far from the expectation.

In the end, this work still requrieds a search of (a wide range of) hyper-parameters. Given the required computation for model training, it can be better if the author(s) can provide more insight on how to choose the hyper-parameters (e.g. rule-of-thumb) instead of through experiments.

---

> ### Author Response · Authors · 2022-08-02
> **Response to Reviewer uRAL (1/2)**
>
> **Q1:** I understand it is tedious to prepare the code and data sets. But I still strongly suggest the authors provide the code and data sets to make the process more transparent.
>
> **A1:** Thank you for the suggestion. We have planned to release the source code and detailed instructions for reproducing our results upon acceptance of this paper (Checklist 3 (a)). However, to avoid potential misunderstandings, we have uploaded the source code to an anonymous link below: https://anonymous.4open.science/r/NeurIPS2022-Paper5388-FreeGEM. Due to time constraints, we only prepared the code which can be used to reproduce the results of Table 3: Accuracy comparison with state-of-the-art methods. The rest of the code will be released in the final version.
>
> The datasets used in our paper are all publicly available, and we have put the links to the datasets in the README file. In addition, since we are targeting at dynamic link prediction problems, i.e., predict future links based on previous links, we split the training, validation and test sets by chronological order so that there is no randomness in the splitting of datasets (Checklist 3 (c)).
>
> **Q2:** I have my reservation related to the content of checklist where the author(s) wrote "To the best of our knowledge, this work does not have potential negative social impacts." I like the confidence signaled. However, when it comes to user-item data analysis, a common ethical issue is that some users and items may suffer because they belong to some under represented groups. Especially when the training data is splited into fragments. I think the paper can be better if the author(s) elaborate more on this direction and futhermore the potentials of this work or continuations of this work can help mediate the mentioned issue.
>
> **A2:** Thank you for the suggestion. We agree that the "under represented groups" you mentioned may be a severe issue in specific applications and the existence of "under represented groups" usually comes from the biases during the data collection process. However, we believe the biases in those datasets will not be explicitly amplified by our method. Especially, in the Online module, we update the user/item embedding for each new user-item interaction, instead of batch updates, so that every user/item is treated equally. After a period of online parameter updates, we restart the Offline module for calibration using truncated SVD. During offline computation, all user-item interactions, instead of batch of interactions, are considered and treated equally.
>
> To further evaluate our method on biased datasets, we have performed a preliminary analysis on ML-100K. First, we group the users by gender and find that there are 670 male users (71.0%) and 273 female users (29.0%) in this dataset. Male users had 753313 interaction records (74.4%), and female users had 246298 interaction records (25.6%). This shows that there is bias of gender distribution in this dataset. Then, we remove the attribute-integrated SVD module from our method and calculate the performance of our model on male and female users. We find that without using attributes, the Recall of male users is 0.124 and Recall of female users is 0.085. Male users are with 45.9% higher Recall than female users. Finally, we take the attribute-integrated SVD module back to see how the performance differs. We find that after using attributes, the Recall of male users increased to 0.156, Recall of female users increased to 0.127. Male users are with 22.8% higher Recall than female users.
>
> To sum up, due to the bias of gender distribution in the ML-100K dataset, there is a clear bias in recommendation accuracy, i.e., male users are with 45.9% higher Recall than female users even without using user attributes in the model. However, to our surprise, the bias in the recommendation results becomes less significant after using user attributes in our model, i.e., male users are with only 22.8% higher Recall than female users when user attributes are incorporated. We think the reason is that user attributes can help these under represented groups (female users) to improve their embedding quality. However, although the bias in the recommendation results is alleviated by our method, it is not completely eliminated. Thus, for practitioners who may apply our method in applications with ethical issues in their datasets, e.g., with under represented groups as mentioned by the reviewer, they should carefully evaluate the predictions made by our method before using them. If there are clear biases in the predictions, they should apply bias mitigation techniques, e.g., [1], in combination with our method. We will present these new results and discussions in the final version.
>
> [1] Shrikant Saxena, Shweta Jain. Exploring and Mitigating Gender Bias in Recommender Systems with Explicit Feedback. arXiv:2112.02530. 2021.

---

> > ### Author Response · Authors · 2022-08-02
> > **Response to Reviewer uRAL (2/2)**
> >
> > **Q3:** The idea of parameter-free is always exciting in data analysis. However, I cannot fully agree with the definition proposed by the author(s). I expect to see a model without or with less complicated hyper-parameter or training parameter set-ups. However, according to the definition:
> >
> > "...we do not incorporate any parameters which need to be learned via back-propagation..." the reality is far from the expectation.
> >
> > In the end, this work still requrieds a search of (a wide range of) hyper-parameters. Given the required computation for model training, it can be better if the author(s) can provide more insight on how to choose the hyper-parameters (e.g. rule-of-thumb) instead of through experiments.
> >
> > **A3:** Thanks for the suggestion. Indeed, as mentioned in the paper, "parameter-free" means that "we do not incorporate any parameters which need to be learned via back-propagation". If the word "parameter-free" may cause potential confusions, we would like to change it to *learning-free* or *training-free* which should be more clear.
> >
> > With regard to hyper-parameters, we have the following observations and will add them in the final version of this paper.
> >
> > (1) Hyper-parameters related to the attribute-integrated SVD module include $k_2,$ $k_3$, $k_4$, $k_5$ and $\alpha_1$ ,$\alpha_2$, $\alpha_3$. Due to the inclusion of attribute information, this module can improve the prediction accuracy but also introduce several more hyper-parameters. It can be observed in ablation experiments that our method can still surpass other methods even without using attribute information. Thus, in practice, attribute information can be used only for those users with scarce interactions or cold start users, which can significantly improve the accuracy as shown in Section 5.4. In addition, during hyper-parameter search, we observe that using $k_2=k_3=k_4=k_5$ can achieve outstanding performance, even though they can be tuned separately.
> >
> > (2) The hyper-parameter $d$ that controls the restart of the Offline module has less significant effect on the prediction accuracy. Its role is to control the restart times according to the data scale. As we can see, in Wikipedia dataset with small data scale, we use the search interval of 5, 10, ..., 50, while in LastFm dataset with large data scale, we use the search interval of 300, 400, ..., 800.
> >
> > (3) In the experiment, we found that for different datasets, the hyper-parameter $\beta$, which control the time decay is sensitive to the dataset. We can set higher priority for the searching of $\beta$. Luckily, $\beta$ is not very sensitive to the values of other hyper-parameters, so that we can search other hyper-parameters after the optimal $\beta$ is found.
> >
> > (4) As shown in Table 6, the model training time of our method is much shorter compared to other methods for one group of hyper-parameters, especially on larger dataset. Thus, we found that the overall running time (include hyper-parameter searching) of our method is still much lower than the other methods. We will report the detailed results in the final version.

---

> ### Author Response · Authors · 2022-08-07
> **Thanks for reading our rebuttal**
>
> Dear Reviewer uRAL,
>
> Once again, we would like to thank you for your constructive comments in your initial reviews. We hope that we have addressed all the concerns raised in your reviews in the rebuttal. Please do let us know if you still have any concern about this paper and we will be more than happy to discuss with you before the discussion period ends.
>
> Best regards,
>
> Authors of Submission 5388

---

> > ### Comment · Reviewer_uRAL · 2022-08-07
> > **OK**
> >
> > Patience :)

---

### Author Response · Authors · 2022-08-02
**General Response to AC and Reviewers**

Dear AC and reviewers,

We would like to thank all of you for your great efforts, insightful comments and detailed suggestions, which are very helpful for us to further improve this paper. Next, we would like to respond to the questions raised by the reviewers one by one, and we will incorporate these clarifications in the final version if this paper is accepted.

Best regards,

Authors of Submission 5388

---

### Meta-Review · Area_Chair_E3mb · 2022-08-27

**Recommendation:** Accept
**Confidence:** Less certain

**Metareview:**

Reviewers were overall on the side of acceptance although not strongly so, with one reviewer being only borderline.

Several positive aspects of the paper were appreciated:
- The idea was considered novel and the motivation clear.
- The combination of matrix factorization with dynamic graph-structured data,
 was appreciated, and the methodological use of the two modules and online-monitor-offline architecture were considered novel.
- The efficiency improvement was appreciated.
- The experiments were considered comprehensive and sufficient.
- The paper was considered well organized.

On the negative side,
- Only addressing link prediction was considered too limited.
- The method was criticized as still requiring search across hyperparameters.
- More clarification of the methodology was desired, and the role of the different submodules compared to functionalities in other methods was considered hard to understand.
- Lack of provided code and data sets was criticized; a partial code release was provided by authors in their response. uRAL
- Potential negative social impacts for underrepresented groups could have been mentioned.

Overall, I think this paper can be acceptable for NeurIPS if the authors take the reviewer comments and the follow-up discussion into account in their final manuscript.

**Award:**

No

---

### Decision · Program_Chairs · 2022-09-14

Accept